# Digital Intraoral Scanners and Alginate Impressions in Reproducing Full Dental Arches: A Comparative 3D Assessment

**Kyungmin Clara Lee** [1,*] and **Seong-Joon Park** [2]

1    Department of Orthodontics, School of Dentistry, Chonnam National University, Gwangju 61186, Korea
2    Barun-e Orthodontic Clinic, Namyangju 472901, Korea; orthopsj@hanmail.net
*    Correspondence: ortholkm@jnu.ac.kr; Tel.: +82-62-530-5864

**Abstract:** The purpose of this in vivo study was to compare in vivo full arch intraoral scans obtained using two intraoral scanners and conventional impression. Twenty patients were scanned using TRIOS and iTero scanners, as well as conventional impression. Dental models obtained from alginate impression were scanned with a laboratory desktop scanner. Individual intraoral scan data were compared with corresponding model scans using three-dimensional (3D) surface analysis. The average surface deviations were calculated for quantitative evaluation, and these values were compared between two intraoral scanners using the paired *t*-test. In the 3D surface analysis, most deviations between intraoral scans and model scans presented on the posterior teeth. The average surface deviations were less than 0.10 ± 0.03 mm. The results of 3D surface analysis indicated that there was 0.10 mm of overall deviation between conventional alginate impressions and in vivo full dental arch intraoral scans. Clinicians should take this into consideration when performing intraoral scanning for full dental arches.

**Keywords:** intraoral scan; conventional impression; digital impression

## 1. Introduction

Conventional alginate impression techniques have been challenging for patients with sensitive gag reflexes, causing feelings of irritation and discomfort [1]. The first steps in computer-aided design/computer-aided manufacturing (CAD/CAM) systems in dentistry were taken by Mörmann and coworkers in 1987 [2]. Further, Dr. Francois Duret introduced CAD/CAM concepts into dentistry in 1989 [3–5], and several intraoral scanners have been introduced recently [6,7]. Imaging technology is now developing, and intraoral scanners have been steadily upgraded. Since the plaster model and intraoral scans are used together in clinics, it is necessary to evaluate the difference between alginate impression and intraoral scanners. In orthodontic clinics, alginate impression is still being used to fabricate orthodontic diagnostic models or orthodontic appliances. In the present study, the two intraoral scanners were used, TRIOS and iTero. The scanning technology of the two scanners is confocal microscopy; however, the light sources of the two scanners are different. TRIOS is based on a structured-light scanner with an infrared light inside, whereas iTero uses laser as its light source.

Since the introduction of digital impressions with intraoral scanners, the advantages of digital impressions and scanning systems have been fixing the shortcomings of conventional impression using a tray and materials [8]. Intraoral scanners can now take full arches, and intraoral scanning is available for both prosthetic and orthodontic diagnosis and treatment plans [9].

The accuracy of intraoral scanners has been evaluated for single teeth [10–16] and short-span fixed dental prostheses [17–20]. In order to determine the accuracy of intraoral scanners, researchers

have performed in vitro studies using reference models [21–26]. Although short-span intraoral scans have exhibited excellent accuracy, few studies have investigated the accuracy of in vivo intraoral scans for full arch scans. The validity of intraoral scans was limited to tooth size measurements [27], and three-dimensional (3D) superimpositions were not performed with conventional impression [28]. While recent studies have compared conventional impression and intraoral scanning [29–31], there are still limitations in their in vitro studies [29,30] and small sample size [30,31]. The purpose of the present study was to compare in vivo full arch intraoral scans and conventional impressions using 3D surface analysis.

## 2. Materials and Methods

Twenty patients (8 men, 12 women; mean age, 25.5 years) were enrolled in the study, and all patients provided informed consent in written form. The present study was approved by the Institutional Review Board of Chonnam National University Dental Hospital (CNUDH-2015-003). Participants with severe crowding, missing teeth, and restorative teeth such as crowns or bridges were excluded. Each patient underwent intraoral scanning with TRIOS 2 (3Shape, Copenhagen, Denmark) and iTero (version 1.0, Align Technology, San Jose, CA, USA) as well as alginate impression. A single examiner performed the intraoral scanning. TRIOS scans started from the left side. After occlusal surfaces were scanned, lingual and buccal surface scans were performed. In the maxillary arch, occlusal surfaces were scanned first, same as in the mandibular arch, whereas buccal and lingual surfaces were scanned in order. The image could be continuously viewed on screen during the scanning, which allowed direct visual feedback to ensure that no areas were missed. All scan data were sent to the OrthoAnalyzer™ (3Shape) software program, and data were reprocessed as a stereolithography (STL) file. The iTero scanning also started from the left side. After scanning, the data were reprocessed as an STL file.

Maxillary and mandibular alginate impressions (Cavex Impressional, Cavex Holland BV, Haarlem, The Netherlands) were taken in a metal tray and immediately poured with dental stone (New Plastone II White, GC Corporation Tokyo, Japan). The alginate impression was immediately poured with dental stone to prevent dimensional shrinkage. The plaster models were scanned with a laboratory desktop scanner (Orapix, Seoul, Korea). By means of a reverse engineering software program (Rapidform 2006, Inus, Seoul, Korea), the laboratory-scanned file was converted to STL file format.

### 2.1. 3D Compare Analysis of Intraoral Scans

Each intraoral scan obtained using iTero and TRIOS scanners was compared with model scans in this study (Figure 1). Intraoral scans were aligned according to best-fit alignment to the corresponding model scan using a software program (Rapidform 2006, Inus). The initial registration involved the selection of three corresponding points on each of the two models, after which the program's automatic fine-registration function was employed to finalize the alignment (Figure 2). The incisal midpoint and mesiopalatal cups of the right second molar and mesiopalatal cups of the left first molar were used as three reference points. Best-fit alignment was performed with an iterative closest point algorithm on the basis of the laboratory scanned model. The deviation between each intraoral scan and model scan at whole surfaces (points cloud) were assessed with a color histogram. For quantitative evaluation, the average surface deviations between each intraoral scan and the model scan were computed automatically at whole surfaces using the "shell-to-shell deviation" function of the software program.

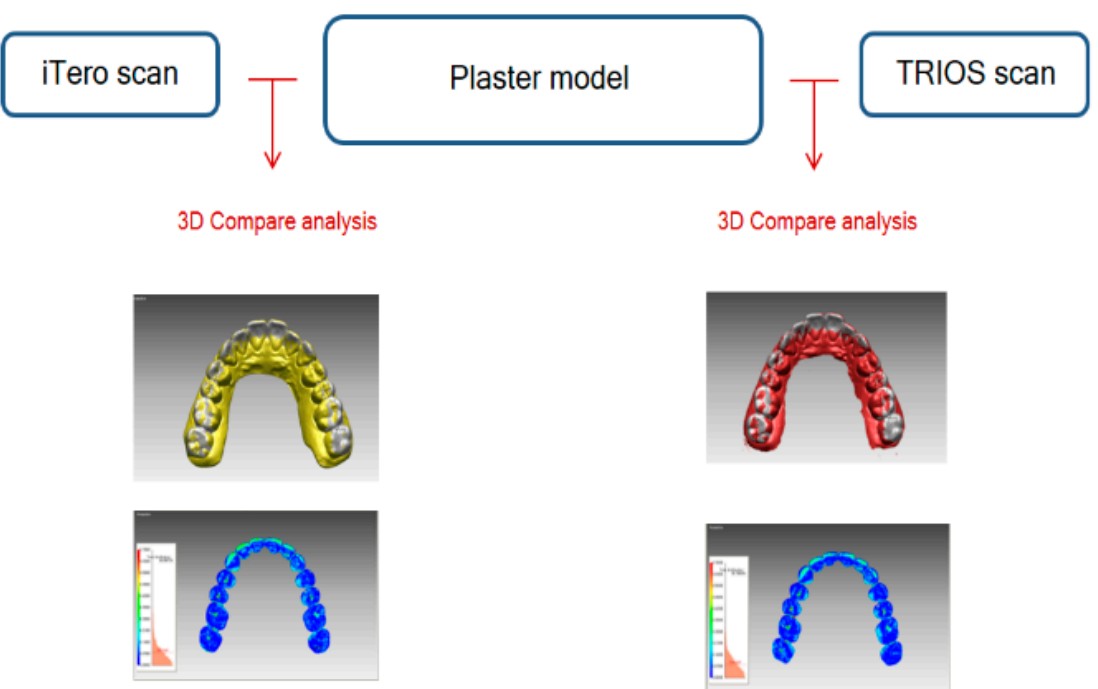

**Figure 1.** Illustration of comparison between intraoral scan and model scan.

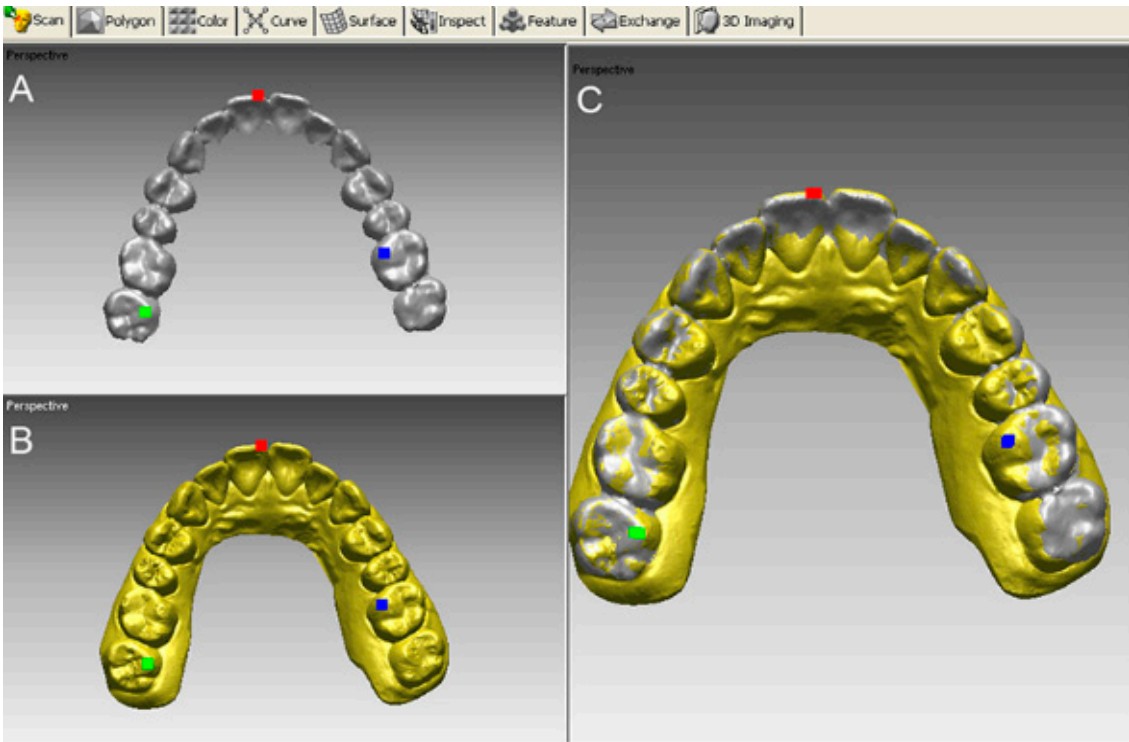

**Figure 2.** An example of best-fit alignment of a laboratory scanner model (**A**) and an iTero scan (**B**) of the maxillary arch. Initial registration of the two scans was achieved by designating three corresponding points. Best-fit alignment was completed using the "fine registration" function of the software program (**C**).

*2.2. Statistical Analysis*

For statistical analysis, the values of shell-to-shell deviation between the TRIOS model and iTero model were used. In order to reveal the difference between the two intraoral scanners, a paired *t*-test was used to compare the values using version 23.0 of the SPSS software package (IBM, Armonk,

NY, USA). The sample size was not calculated a priori; however, power analysis using the G*Power program showed over 90% power for the measurement. For the reproducibility of measurements, all processes including intraoral scanning and its 3D surface analysis were repeated for 10 randomly selected participants, and the magnitude of that error was assessed by calculating the intraclass correlation coefficient (ICC).

## 3. Results

ICC values, showing a range of 0.741 to ~0.924, indicated moderate reproducibility. Color histograms show that the deviations between the intraoral scan and corresponding model scan occurred on the posterior areas (Figure 3). Table 1 shows the means and standard deviations for the average surface difference between each intraoral scan and the model scan. The average surface difference was less than 0.10 mm in both the maxilla and the mandible. No statistically significant differences were found between the TRIOS and iTero scans (Table 1).

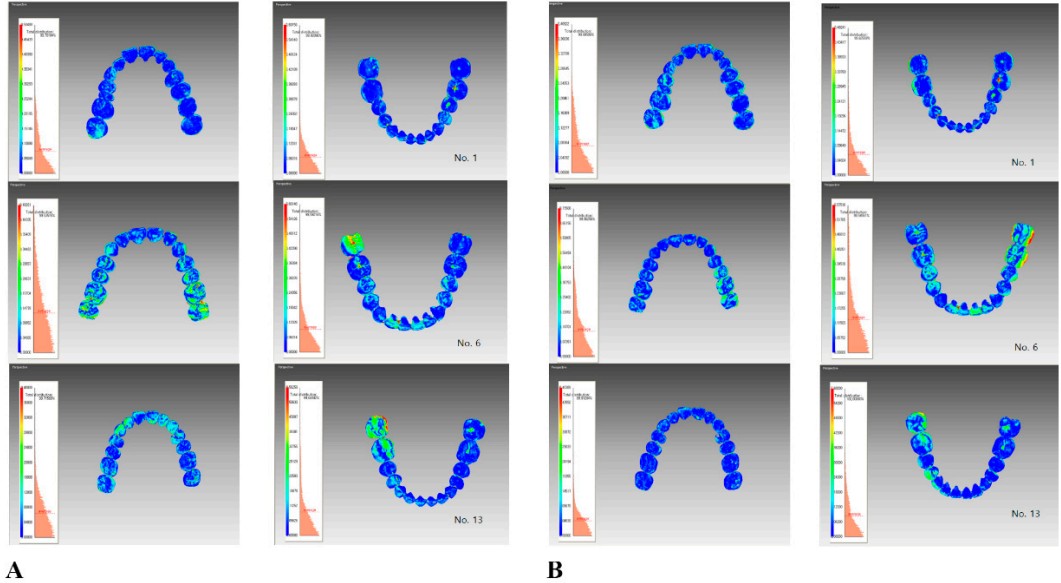

**Figure 3.** Color histograms showing the deviations between the intraoral scan and models after best-fit alignment. (**A**), Surface deviation between the iTero scan and model scan. (**B**), Surface deviation between the TRIOS scan and model scan.

**Table 1.** Surface deviations between each intraoral scan and model scan (unit: mm).

| | iTero **vs.** *Model Scan* | TRIOS **vs.** *Model Scan* | *p* **Value** |
|---|---|---|---|
| | *Mean* ± *SD* | *Mean* ± *SD* | |
| Maxilla | 0.10 ± 0.03 | 0.09 ± 0.03 | 0.206 |
| Mandible | 0.09 ± 0.02 | 0.09 ± 0.03 | 0.359 |

*SD*, Standard deviation.

## 4. Discussion

The aim of this study was to compare conventional impression and digital impression using intraoral scanners qualitatively; 3D surface analysis was a single measurement. Thus, the presence of random errors has been neglected in the present results. In the present results, the average deviations between each intraoral scan and the model scan were within 0.10 mm. The acceptance level of deviation is different on its use. For instance, a 0.12 mm margin discrepancy has been reported to be the limit for a clinically acceptable crown margin in prosthetics [32]. For orthodontics, Hirogaki et al. [33]. suggested that study models' accuracy should be about 0.30 mm, while Schirmer and Wiltshire [34] reported

that a measurement difference of less than 0.20 mm was clinically acceptable. Bell et al. [35] suggested that a measurement difference within 0.27 mm was clinically insignificant. However, in the case of fabricating orthodontic appliances, the value of 0.1 mm might be a problem of ill fitness of appliance. Fabricating orthodontic appliances using intraoral scans includes the 3D printing process of appliances. Thus, the potential errors might be increased. The clinically acceptable values reported in previous studies are necessary to reconsider in cases of fabricating and printing appliances using intraoral scans.

After registration of the iTero scan onto the model, the color histogram showed local deviations in the mandibular posterior regions. This finding was consistent with the result of a previous study by Patzelt et al. [24]. The authors reported that most data presented some deviations in the posterior areas [24]. Nedelcu et al. [31]. reported local deviations along the palatal surfaces of the molars and incisal edges of the anterior teeth less than 100 μm. Ender et al. [36]. found that conventional and digital impression methods differed significantly in the complete-arch precision, and digital impression systems had higher local deviations within the complete arch cast. Zimmermann et al. [37]. reported that conventional alginate impression showed the lowest precision compared with intraoral scanning systems (Cerec Omnicam Ortho; Ormco Lythos) with guided scanning procedures for higher precision. As digital impression systems continue to improve and develop rapidly, they may prove to be an equivalent or better alternative to conventional impression techniques.

To evaluate accuracy, two factors are usually considered; trueness and precision. Trueness is the deviation of the impression geometry from the original geometry, while precision is the deviation between repeated impressions [37,38]. In this study, precision between the conventional and intraoral scans was not evaluated; only trueness was evaluated. Regarding trueness, the model scan was considered a reference in this study because the models are still used in orthodontic clinics. Thus, it is necessary to evaluate the difference between alginate impression and the intraoral scanner. To be exact, this study investigated the agreement between the alginate impression and intraoral scans, not the accuracy thereof. In the results, the reason for the deviation between alginate impression and the intraoral scanner could be explained by the scanning process. Furthermore, the scanning accuracy of an in vivo full arch scan is affected by several factors of intraoral conditions. Anatomical limitations due to limited mouth opening and tongue movement might contribute to difficult access to the mandibular posterior areas. Flugge et al. [39]. found that intraoral scanning was less precise than extraoral model scanning, indicating that the intraoral conditions contribute to the scanning inaccuracies. The authors also observed that the deviations occurred at the buccal side of the lower molars and the anterior teeth.

The limitations of this study were as follows: In the study, only 3D surface analysis was used. Specifically, certain areas which had deviations needed to be evaluated. With the rapid development of intraoral scanners, there are other advanced intraoral scanners in the market. In the study, only iTero and Trios scanners were evaluated. In addition, the intraoral scanners used in the study were old models, and this could be related with the accuracy obtained.

## 5. Conclusions

Taking the limitations of this study into account, the results of 3D surface analysis indicated that there was 0.10 mm of overall deviation between conventional alginate impressions and in vivo full dental arch intraoral scans. Clinicians should take this into consideration when performing intraoral scanning for full dental arches. When in vivo full arch scanning is performed, the factors causing scanning errors, such as strong light, irregular calibration of scanner, and too much saliva and moisture in patients' mouth, should be prevented before scanning.

**Author Contributions:** Conceptualization, K.C.L.; methodology, K.C.L.; software, S.-J.P.; validation, K.C.L.; investigation, S.-J.P.; writing—original draft preparation, K.C.L.; writing—review and editing, K.C.L.; supervision, K.C.L.; funding acquisition, K.C.L. All authors have read and agreed to the published version of the manuscript.

**Funding:** This work was supported by the National Research Foundation of Korea (NRF) grant funded by the Korean government (Ministry of Science and ICT) (No. 2020R1F1A1070617 and NRF-2017R1D1A1B03032132). All the scanners and materials used here belonged to the author, and nothing was provided by third parties or private companies; therefore, the authors have no conflict of interest related to the present work.

**Conflicts of Interest:** The authors declare no conflict of interest.

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
