# Peer review of "Digital Intraoral Scanners and Alginate Impressions in Reproducing Full Dental Arches: A Comparative 3D Assessment"

_applsci, doi:10.3390/app10217637_

Round 1
Reviewer 1 Report
I reviewed the present study that digitally compares full dental arches models obtained by alginate impressions and digital scanners.
Even if not completely original the study can improve the knowledge in the field thus could be acceptable for publication.
I provide some suggestions/concerns about the manuscript:
- Did you check the data distribution before applying a parametric test?
- I think that paired t-test is not the ideal statistical analysis considering that the deviations between alginate and digital impressions obtained with two different scanners are not considerable as repeated measurements but as independent ones. I thus, suggest to apply independent samples t-test.
- Please at the end of the discussion section add some considerations about the limitations of the present study.
- In the conclusions section explain why clinician should take into consideration the results of the present study.
Author Response
I reviewed the present study that digitally compares full dental arches models obtained by alginate impressions and digital scanners.
Even if not completely original the study can improve the knowledge in the field thus could be acceptable for publication. I provide some suggestions/concerns about the manuscript:
- Did you check the data distribution before applying a parametric test?
- Response 1. Normality test was performed.
2.I think that paired t-test is not the ideal statistical analysis considering that the deviations between alginate and digital impressions obtained with two different scanners are not considerable as repeated measurements but as independent ones. I thus, suggest to apply independent samples t-test.
Response 2. We compared the results of dental models corresponding alginate impressions, thus, paired t-test was done.
- Please at the end of the discussion section add some considerations about the limitations of the present study.
Response 3. We added the limitations of the present study as follows; The limitations of this study were as follows; In the study, only 3D surface analysis was used. Specifically, certain areas which have deviations were need to evaluated. With the rapid development of intraoral scanners, there are other advanced intraoral scanners in the market. In the study only iTero and Trios scanners were evaluated.
4.In the conclusions section explain why clinician should take into consideration the results of the present study.
Response 4. We added the text in conclusion as follows; With the limitation of this study, the results of 3D surface analysis indicated that there was 0.10 mm of overall deviation between conventional alginate impressions and in vivo full dental arch intraoral scans. Clinicians should take into consideration of this when performing intraoral scanning for full dental arches. When the in vivo full arch scanning is performed, the factors causing the scanning errors such as strong light, irregular calibration of scanner, too much of saliva and moisture in patients’ mouth can be prevented before scanning.
Reviewer 2 Report
The manuscript has been improved, although some points are still to be mentioned.
Some grammar mistakes can be found throughout the manuscript. Please re-check for proper Enlish language again.
Materials and methods:
p2, 2nd par: please provide some information about the tooth surface area which was selected (1) for best fit alignment and (2) for deviation analyses. Which part of the teeth was taken into account?
p2, 3rd par: statistics: why was a paired t-test used? Was the dataset normally distributed?
Results:
ICC values of 0.75-0.92 are at most sufficient but seem not "excellent" to me.
The discussion section has been fundamentally improved.
Author Response
The manuscript has been improved, although some points are still to be mentioned.
Some grammar mistakes can be found throughout the manuscript. Please re-check for proper Enlish language again.
Response 1. Thank you for your comments. We have checked the English language one more time.
Materials and methods:
p2, 2nd par: please provide some information about the tooth surface area which was selected (1) for best fit alignment and (2) for deviation analyses. Which part of the teeth was taken into account?
Response 2. The incisal midpoint and mesiopalatal cups of right second molar and mesiopalatal cups of left first molar were selected for best fit alignment and after that which the program’s automatic fine-registration function was employed to finalize the alignment. For deviation analysis, whole surfaces were used.
p2, 3rd par: statistics: why was a paired t-test used? Was the dataset normally distributed?
Response 3. Normality test was performed. We compared the results of dental models corresponding alginate impressions, thus, paired t-test was done.
Results:
ICC values of 0.75-0.92 are at most sufficient but seem not "excellent" to me.
Response 4. We have revised the sentence about that according to your comments as follow; ICC values, showing a range of 0.741 ~ 0.924, indicated moderate reproducibility.
The discussion section has been fundamentally improved.
Response 5. Thank you for your comment.
Round 2
Reviewer 1 Report
The paper is now almost ready to be accepted.
I suggest to add to the limitations also that the scanners considered in the study are also old models and this could be related with the accuracy obtained.
Best regards
Author Response
The paper is now almost ready to be accepted. I suggest to add to the limitations also that the scanners considered in the study are also old models and this could be related with the accuracy obtained.
[response] Thank you for your thoughtful comment. We have added this sentence according to your comment as follows; In addition, the intraoral scanners used in the study were old models and this could be related with the accuracy obtained.
Reviewer 2 Report
Thank you very much for the revised manuscript, I have no further points to note.
Author Response
Thank you very much for the revised manuscript, I have no further points to note.
[Response] Thank you for your comment.
This manuscript is a resubmission of an earlier submission. The following is a list of the peer review reports and author responses from that submission.
Round 1
Reviewer 1 Report
I really wonder what new aspects this paper tells us. It is long known that the precision of intraoral scans compared to gypsum models is comparably good. Furthermore, there are a lot of shortcomings
The real interesting question, namely the trueness of the scans is not addressed.
Additionally, solely from the point of the literature cited the paper is completely inadequate. There are two citations from 2017 and not a single one more recent. This is an absolute no go in this rapidly developing field.
- There is no clear description of the conventional impression
- Who did the scans?
- Which scanner version of the itero / trios was used?
- Positive and negative deviations were analyzed together. As they eliminate each other a scientifically sound paper needs a separate analysis of the positive and negative values.
- Obviously only one scan was used for measurement?? à Our intention was to compare the conventional impression and intraoral scanners qualitatively; 3D surface analysis was used for one single measurement. ???
- Thus the results presented are more than sparse
- Technically the section discussion is missing as there is no headline
- I guess discussion starts at l 117
- L118 to 139 are no discussion, at best introduction, a repetition of the results and the first half of the paragraph can be expected to be standing knowledge in the scientific community. This part can be omitted.
- The discussion does not reflect the method, overall the remaining discussion is more than sparse and we learn nothing new.
The English needs decisive improvement
Author Response
1.There is no clear description of the conventional impression
Response 1. Alginate impression was used as conventional impression. We described the detail in the materials.
2.Who did the scans?
Response 2. Single examiner performed intraroal scan.
3.Which scanner version of the itero / trios was used?
Response 3. Itero version 1.0 and Trios 2 were used.
4.Positive and negative deviations were analyzed together. As they eliminate each other a scientifically sound paper needs a separate analysis of the positive and negative values.
Response 4. The value used in the study is shell/shell deviation and this is always positive value.
5.Obviously only one scan was used for measurement?? à Our intention was to compare the conventional impression and intraoral scanners qualitatively; 3D surface analysis was used for one single measurement. ???
Response 5. This means that the result of 3D surface analysis consists of single value.
6.Thus the results presented are more than sparse
Response 6. We revised the results according to your comment.
7.Technically the section discussion is missing as there is no headline
I guess discussion starts at l 117. L118 to 139 are no discussion, at best introduction, a repetition of the results and the first half of the paragraph can be expected to be standing knowledge in the scientific community. This part can be omitted.
Response 7. The section title was added ad discussion was corrected according to your comment.
8.The discussion does not reflect the method, overall the remaining discussion is more than sparse and we learn nothing new.
Response 8. We revised the discussion as much as possible.
9.The English needs decisive improvement.
Response 9. The manuscript was checked once again through professional institution of English editing system.
Reviewer 2 Report
This manuscript is well made, but I think that Alginate use , also in orthodontic , have limited space of utilization
Author Response
This manuscript is well made, but I think that Alginate use, also in orthodontic, have limited space of utilization.
Response 1. We totally agree with your opinion. However, alginate is still commonly used in clinics including orthodontic clinics. We tried to simulate the clinical situation which use alginate impressions for the purpose of provisional diagnosis.
Reviewer 3 Report
Dear Authors, I have carefully read the submitted manuscript. Even though the topic is interesting and attractive, the study design and the manuscript presentation make it not acceptable for publication. Indeed, the background appear old and not updated, the methods are not appropriate and not well described.
I strongly recommend the rejection of the manuscript.
Author Response
Response 1. This study compared the in-vivo full arch intraoral scans obtained using two commercially available intraoral scanners and conventional impression using 3D surface analysis. Thank you for your comments.
Reviewer 4 Report
I reviewed the present study that digitally compares full dental arches models obtained by alginate impressions and digital scanners.
Even if not completely original the study is well designed and can improve the knowledge in the field thus could be acceptable for publication. Anyway, the manuscript is actually poor and must be improved before being accepted. A Major revision is necessary and I will provide my point by point suggestions:
- The title in my opinion is not appropriate for introducing a scientific study, sounds more like a literature review or a commentary on this topic. I suggest something more similar to “Digital intraoral scanners and alginate impressions in reproducing full dental arches: A comparative 3D assessment.”
- In the introduction and discussion the authors try to explain that this kind of comparison was done on the basis that alginate impressions are commonly used for study models fabrication in the orthodontic practice, but in the discussion should be analyzed the importance of the possible differences in the impact on study model analysis or orthodontic appliances contruction.
- Furthermore, the comparison was performed between the two scanners instead than between the scanners and the alginate derived models… this statistical analysis should be added if one of the objectives of the study is understanding is digital models could substitute alginate impressions in the dental/orthodontic practice. Were the differences between the intraoral scan and alginate impressions statistically significant? Are the same alginate models accurate (as described by the previous literature)?
- The literature background must be improved in introduction and even more in the discussion section (even if many citations are in the reference list!). Too many studies not directly comparable or related with the present one are cited while many comparable study were forgotten (ex. Accuracy of digital models generated by conventional impression/plaster-model methods and intraoral scanning. Dent Mater J. 2018 Jul 29;37(4):628-633.; Comparison of Accuracy Between a Conventional and Two Digital Intraoral Impression Techniques.Int J Prosthodont. 2018 Mar/Apr;31(2):107-113.) Please check and improve the literature background.
- In the methods section is not specified which edition of TRIOS and iTero were used in the present study. This is very important as innovations were introduced by the producers in both scanners over time.
- How the 3 points for superimposition were chosed?
- The Table in page 4 is without appropriate number… it should be called “Table 1”
- There is no discussion section title… I suppose it should be placed after the table in page 4.
- As already said before the discussion is poor and must be improved with other comparisons with the previous literature results in similar studies and expressing the clinical implications of the findings of the present study.
- All the pronouns as “our” should be removed from the manuscript and the third person should be constantly used speaking about the study.
- The conclusions section is too small and must be improved with a couple of sentences that reflects what could be concluded about the aim of the study and the clinical implications.
Author Response
- The title in my opinion is not appropriate for introducing a scientific study, sounds more like a literature review or a commentary on this topic. I suggest something more similar to “Digital intraoral scanners and alginate impressions in reproducing full dental arches: A comparative 3D assessment.”
Response 1. Thank you for your suggestion. We changed the title according to your suggestion.
- In the introduction and discussion the authors try to explain that this kind of comparison was done on the basis that alginate impressions are commonly used for study models fabrication in the orthodontic practice, but in the discussion should be analyzed the importance of the possible differences in the impact on study model analysis or orthodontic appliances contruction.
Response 2. We have revised the discussion according to your comments.
- Furthermore, the comparison was performed between the two scanners instead than between the scanners and the alginate derived models… this statistical analysis should be added if one of the objectives of the study is understanding is digital models could substitute alginate impressions in the dental/orthodontic practice. Were the differences between the intraoral scan and alginate impressions statistically significant? Are the same alginate models accurate (as described by the previous literature)?
Response 3. By nature of 3D surface analysis, there is only comparison value between alginate impression and intraoral scan, no each measurement in alginate impression and intraoral scan. Thus there is no statistical analysis between alginate impression and intraoral scan. We can only evaluate the comparison value using 3D surface analysis, particularly color map.
- The literature background must be improved in introduction and even more in the discussion section (even if many citations are in the reference list!). Too many studies not directly comparable or related with the present one are cited while many comparable study were forgotten (ex. Accuracy of digital models generated by conventional impression/plaster-model methods and intraoral scanning. Dent Mater J. 2018 Jul 29;37(4):628-633.; Comparison of Accuracy Between a Conventional and Two Digital Intraoral Impression Techniques.Int J Prosthodont. 2018 Mar/Apr;31(2):107-113.) Please check and improve the literature background.
Response 4. Thank you for your suggestions. We improved the introduction with adding these references.
- In the methods section is not specified which edition of TRIOS and iTero were used in the present study. This is very important as innovations were introduced by the producers in both scanners over time.
Response 5. TRIOS version 2 and iTero 1 were used in this study. We added the text of this in the manuscript.
- How the 3 points for superimposition were chosed?
Response 6. The incisal midpoint and mesiopalatal cups of right second molar and mesiopalatal cups of left first molar were used as three reference points. We added the text of this in the manuscript.
- The Table in page 4 is without appropriate number… it should be called “Table 1”
Response 7. We corrected it.
- There is no discussion section title… I suppose it should be placed after the table in page 4.
Response 8. We wrote the section title.
- As already said before the discussion is poor and must be improved with other comparisons with the previous literature results in similar studies and expressing the clinical implications of the findings of the present study.
Response 9. Thank you for your suggestions. We improved the Discussion with adding these references.
- All the pronouns as “our” should be removed from the manuscript and the third person should be constantly used speaking about the study.
Response 10. We rephrased this through the manuscript.
- The conclusions section is too small and must be improved with a couple of sentences that reflects what could be concluded about the aim of the study and the clinical implications.
Response 11. We revised the conclusion according to your comment.
Reviewer 5 Report
The manuscript presents an investigation of the accurary of intraoral scanning devices.
General remarks: The language needs a careful revision, as the manuscript is full of grammatical errors, which make it difficult to understand.
The study protocol includes alginate impressions as reference for intraoral scanning devices. These type of impression in far to imprecise. A much more precise type of impression (polyether, silicone) is required for such measurements.
From my opinion, the mentioned 120 µm margin discrepancy is far to high, as no sufficient adaptation of the restoration is possible with such a fit in this crucial area.
Therefore, this protocol is not sufficient for investigating this aspect of digital dentistry from my point of view.
The selection of the references has to be improved on some points, for examle on p. 4 ll.118: The first steps of CAD/CAM-systems in dentistry were made by Mörmann and coworkers, even 2 years earlier (1987).
Further, the conclusions drawn by the authors are not supported by the presented data from this investigation.
Author Response
The manuscript presents an investigation of the accurary of intraoral scanning devices. General remarks: The language needs a careful revision, as the manuscript is full of grammatical errors, which make it difficult to understand.
Response 1. According to your comment, the manuscript was checked once again through professional institution of English editing system.
The study protocol includes alginate impressions as reference for intraoral scanning devices. These type of impression in far to imprecise. A much more precise type of impression (polyether, silicone) is required for such measurements.
Response 2. We totally agree with your opinion. However, alginate is still commonly used in clinics including orthodontic clinics. We tried to simulate the clinical situation which use alginate impressions for the purpose of provisional diagnosis.
From my opinion, the mentioned 120 µm margin discrepancy is far to high, as no sufficient adaptation of the restoration is possible with such a fit in this crucial area.
Therefore, this protocol is not sufficient for investigating this aspect of digital dentistry from my point of view.
Response 3. We understand your comments and totally agree with your opinion. Our intention of this sentence is the clinically acceptable limit is different according to the purpose of digital scanning. The orthodontic purpose of digital scanning is more acceptable than the prosthetic purposes.
The selection of the references has to be improved on some points, for examle on p. 4 ll.118: The first steps of CAD/CAM-systems in dentistry were made by Mörmann and coworkers, even 2 years earlier (1987).
Response 4. Thank you for your thoughtful comments. We have revised the references according to your comment.
Further, the conclusions drawn by the authors are not supported by the presented data from this investigation.
Response 5. We revised the conclusion to be supported by the results.
Round 2
Reviewer 1 Report
Thank you for the revision. However, from the scientific point of view decisive problems remain:
- In the field of intraoral scanning, there is a lot of recent research available. Provided the latest literature cites from 2018 is completely inappropriate as you omit up to date material. I just checked pubmed and retrieve close to 10 papers that address the full arch topic only in 2020!! These should be reported as well in the intro as the discussion. Old stuff that is known to every expert in this field and does not provide any new knowledge and does not prepare the ground for the topic investigated is inappropriate for a scientific paper.
- What do you mean with the sentence in l. 49 what do you mean with the validity of intraoral scans. ?
- Furthermoree there are no references for l. 38 Trios structured light and iTero Laser .
- In l. 52 you mention that the problem of in vivo studies is their small number of subject. However, your study also includes only 20 subjects. This is also not much.
- The description of the conventional impression is still inappropriate. What do you mean with immediately poured ? Specially for alginates a precise time should be observed. One the one hand the alginate has to be given time to reset and on the other it must not to be stored to long in order to avoid shrinkage. When did you meausure ? How long were the models stored ? How were they stored ? All this remains open though it is essential for the dimension of the plaster.
- The software versions used are not reported.
- May be I do not understand your assessment technique. However, from its principle a best fit analysis always delivers positive and negative values. Otherwise, it is not a best fit analysis. If you have only positive deviations you do not have a best fit ! That is impossible.
- Furthermore, I really do not understand how you exactly determined the reference points. All tooth structures are curves surfaces. A reproducibly determination without additional reference or a coordinate system is good guess but not more – especially when reporting dimensions of about 90 µm!!
- Furthermore, the fact that for nearly all results you 90 µm and nearly an identical std suggest that your results only represent the uncertainty of the method which neither been investigated nor otherwise addressed.
- The single figures in fig 3 are not legible
- I do not understand why a deviation of 0.1 mm over the entire jaw! Should be a problem in orthodontics (l 133 new text). You do not give a reference for this. Furthermore, in the lines above you mention that more than 0.2 mm are now problem.
- This is even more questionable as later on you cite Zimmermann reporting that alginate impressions deliver a lower “precision” than IOS. (I think you mean “trueness” ). This said the trueness of IOS – which you never assessed – is not a limiting factor when manufacturing orthodontic appliances as this works – as we know from clinical work – well with alginates since decades
- How does this fit to l. 161 in your discussion.
- What does your conclusion tell us ? What shall clinicians concider ?
Reviewer 3 Report
The manuscript presentation has been improved, but the scientific method of evaluation used, and the discussion of the results have not changed. As I stated previously, I disagree with the objective and scientific method used in this investigation. The results obtained and presented have a low significance for the field of dentistry.
Reviewer 4 Report
There was an effort of the authors in revising the manuscript.
Almost all the suggestions were followed.
In order to make the article ready to be published I think that should be added a discussion about alginate models accuracy (as described by the previous literature), considering that they were taken as ideal reference. In the present study a discrepancy of 0.1mm was found with alginate impression but is the accuracy of these impressions higher than 0.1mm? Otherwise it cannot be used as a reference for discrepancy lower than 0.1mm. Please check it in the previous literature and add it to the discussion.
Furthermore, the two analyzed intraoral scanners should be removed from the introduction and be mentioned only starting from the methods section, in the introduction I suggest to generally speak about different types of intraoral scanners.
Reviewer 5 Report
As I stated before, I do not agree with the study protocol used in this investigation. From my point of view, the benefit of a comparison of 2 methods only makes sense, if they are referred to a gold standard (e.g. deviation of alginate impressions/desktop scanner to gold standard 100µm // deviation of intraoral scan to gold standard 80µm).
In this case, the two values could be statistically evaluated and compared with each other to investigate the trueness of those procedures.
As this is not addressed by this manuscript, there is only low significance of the presented content in this field of dentistry.